# Patient Nutrition and Probiotic Therapy in COVID-19: What Do We Know in 2021?

**DOI:** 10.3390/nu13103385

**Published:** 2021-09-26

**Authors:** Viktoria Hawryłkowicz, Danuta Lietz-Kijak, Karolina Kaźmierczak-Siedlecka, Joanna Sołek-Pastuszka, Laura Stachowska, Marcin Folwarski, Miłosz Parczewski, Ewa Stachowska

**Affiliations:** 1Department of Human Nutrition and Metabolomics, Pomeranian Medical University, 71-460 Szczecin, Poland; vik.hawrylkowicz@gmail.com (V.H.); laura.stachowska@pum.edu.pl (L.S.); 2Department of Propedeutics, Physiodiagnostics and Dental Physiotherapy, Pomeranian Medical University, 70-111 Szczecin, Poland; zpropst@pum.edu.pl; 3Department of Surgical Oncology, Medical University of Gdansk, 80-214 Gdansk, Poland; leokadia@gumed.edu.pl; 4Department of Anaesthesiology and Intensive Care, Pomeranian Medical University, 71-242 Szczecin, Poland; joanna.pastuszka@pum.edu.pl; 5Department of Clinical Nutrition, Medical University of Gdansk, 80-211 Gdansk, Poland; marcin.folwarski@gumed.edu.pl; 6Department of Infectious, Tropical and Acquired Immunological Diseases, Pomeranian Medical University, 71-455 Szczecin, Poland; mparczewski@yahoo.co.uk

**Keywords:** probiotics, COVID-19, SARS-CoV-2, microbiota, nutrition

## Abstract

Background: The main nutritional consequences of COVID-19 include reduced food intake, hypercatabolism, and rapid muscle wasting. Some studies showed that malnutrition is a significant problem among patients hospitalized due to COVID-19 infection, and the outcome of patients with SARS-CoV-2 is strongly associated with their nutritional status. The purpose of this study was to collect useful information about the possible elements of nutritional and probiotic therapy in patients infected with the SARS-CoV-2 virus. Methods: A narrative review of the literature, including studies published up to 13 September 2021. Results: Probiotics may support patients by inhibiting the ACE2 receptor, i.e., the passage of the virus into the cell, and may also be effective in suppressing the immune response caused by the proinflammatory cytokine cascade. In patients’ diet, it is crucial to ensure an adequate intake of micronutrients, such as omega-3 fatty acids (at 2–4 g/d), selenium (300–450 μg/d) and zinc (30–50 mg/d), and vitamins A (900–700 µg/d), E (135 mg/d), D (20,000–50,000 IU), C (1–2 g/d), B6, and B12. Moreover, the daily calorie intake should amount to ≥1500–2000 with 75–100 g of protein. Conclusion: In conclusion, the treatment of gut dysbiosis involving an adequate intake of prebiotic dietary fiber and probiotics could turn out to be an immensely helpful instrument for immunomodulation, both in COVID-19 patients and prophylactically in individuals with no history of infection.

## 1. Introduction

The latest data from the World Health Organization (WHO, 13 September 2021) show that 224,511,226 people have tested positive for SARS-CoV-2 globally. Unfortunately, 4,627,540 deaths associated with COVID-19 infection were also reported around the world [1]. Prognostic models and initial epidemiological data show that new variants of SARS-CoV-2 are associated with higher transmissibility [2,3]. On the other hand, the number of vaccinated individuals is rapidly growing. WHO reports that 5,776,127,976 vaccine doses have been administered worldwide [1]. It can be expected that the aforementioned virus transmits less efficiently during the warmer months. However, the mechanisms regarding this aspect have not been well investigated. Moreover, currently, SARS-CoV-2 should not be assessed as a seasonal entity [4].

Nutritional status impacts infectious diseases through multiple mechanisms. Among others, overweight/obesity contributes to the overproduction of proinflammatory IL-6 and reduces cytotoxicity of natural killer cells. A worse outcome was observed in obese patients with H1N1 influenza [5]. Malnutrition is also associated with a decrease in protein stores and affects the immune system. Both obesity and malnutrition have an impact on the outcome of COVID-19 [6]. Below, we discuss the link between nutritional status and COVID-19-related aspects. As summarized in Ochoa’s review, three main phenotypes have been identified as having an increased risk of severe infection: older and frail, polymorbid or with a chronic illness, and severely obese [7]. Despite the fact that most infections are asymptomatic or mild, more and more data show possible long-term consequences of the disease [8]. Even in low-risk patients, chronic symptoms of COVID-19 may be observed. In the COVERSCAN study, 42% of adults who had recovered from COVID-19 and were followed up after at least 4 weeks had 10 or more symptoms, and 60% had severe post-COVID-19 syndrome with single-organ (70%) and multiorgan (29%) impairment [9]. In another study, 47,780 patients were monitored after hospital stay due to COVID-19. The mean follow-up time was 140 days. Precisely 29.4% required readmission (mainly due to diabetes, cardiovascular event, chronic kidney, or liver disease), and 12.3% died after the discharge [10].

Some studies showed that malnutrition is a significant problem among patients hospitalized due to COVID-19 infection. In Yu’s study on the Wuhan population, malnutrition graded according to the GLIM criteria [11] was associated with longer hospital stay [12]. Zhao’s data showed that in the ICU, an NRS 2002 score over 3 was correlated with a higher mortality rate and prolonged hospital stay [13]. Nutritional assessment with mNUTRIC score was also proven to predict the outcome of patients with a critical illness due to COVID-19 [14]. Consequently, nutritional screening in the ambulatory setting and during hospitalization is acknowledged as an important issue [15]. Moreover, some markers of inflammation and malnutrition are proven to be helpful for mortality prediction. In Zinellu’s meta-analysis, it was shown that low-serum prealbumin levels were associated with severity and mortality of COVID-19 patients [16]. Several potential effects of micronutrients in supporting the treatment of COVID were also studied [17]. Analyses showed that vitamin D deficiency was found in more than two-thirds of admitted patients, and 42% were selenium deficient [18].

The initial phase of the pandemic forced the introduction of new guidelines on medical treatment. The European Society for Clinical Nutrition and Metabolism (ESPEN) published recommendations on clinical nutrition in light of the COVID-19 pandemic [19]. More than one year of the pandemic crisis worldwide has exposed many scientific questions and future strategies for researchers [20], as well as new perspectives for practical challenges of nutritional support. The review by Thibault underlined some feedback “from the field” of the pandemic war, confronting the guidelines in the practical setting [21]. The proposed algorithm suggested that enteral nutrition (EN) may not always be feasible in the intensive care unit (ICU), especially for patients requiring high-volume oxygen support, and total or supplementary parenteral nutrition (PN) support may be useful. Additionally, ease of use and feasibility of nutrition protocols are essential, considering the shortage of healthcare professionals. The authors of the report suggest that measurement of energy expenditure may not always be possible with indirect calorimetry due to problems with the decontamination of equipment, and predictive equations may be helpful in such situations.

Guidelines for patients who recovered from infection stress the need for nutritional surveillance and community support [22]. Although SARS-CoV-2 infection is mainly associated with respiratory problems, a retrospective analysis of 1141 cases of COVID-19 patients showed that 16% had only gastrointestinal (GI) symptoms [23]. A recent meta-analysis confirmed that 11.8% with mild and 17.1% with a severe course of infection had GI symptoms [24].

Notably, the link between gut dysbiosis and respiratory tract infections has been observed [25]. The potential gut–lung axis is regarded as one of the target mechanisms to control the infection and support treatment. Some patients with COVID-19 present decreased levels of probiotic bacteria, such as *Lactobacillus* and *Bifidobacterium*, in the gut [26]. Zuo et al. have shown that fecal metagenomes of high SARS-CoV-2 infectivity patients are characterized by the abundance of opportunistic pathogens (*Collinsella aerofaciens*, *Collinsella tanakaei*, *Streptococcus infantis*, and *Morganella morganii*) [27]. These bacteria affect the immune system both locally and systemically [28]. Microbiota modulation is a promising strategy to improve immunological mechanisms and limit the effect of viral infections [29,30]. Probiotic strains increase regulatory T cells, decrease the production of proinflammatory cytokines, increase the release of anti-inflammatory mediators, and ameliorate antiviral defense [27,28]. Moreover, probiotics enhance mucosal immunity and improve both the intestinal and lung barrier, helping to maintain homeostasis [27,31]. Probiotics enhance the immune system and reduce the inflammatory status then may contribute to act against SARS-CoV-2 [32,33]. In addition, a potential probiotic-induced inhibitory effect on the angiotensin-converting enzyme 2 (ACE2) has been hypothesized [27,34]. The modification of ACE2 expression via microbes has an impact on the entrance of SARS-CoV-2 into cells [34]. Notably, ACE2 is highly expressed in the pulmonary and gut tissues. The spike protein of the aforementioned virus targets ACE2 as the binding receptor for cell entry. Probiotics that are given orally provide antiviral effects through the gut–lung axis [35]. Gut microbiota may modulate the immunological activity of the lung through bacterial metabolites and bacterial lipopolysaccharides [36]. Notably, bacterial metabolites, such as short-chain fatty acids, lipopolysaccharides, and exopolysaccharides, provide an indirect antiviral effect via modulation of the host immune system (the impact on IFN-γ, IgA, IL-12, NK cells, and many others) [37]. The human microbiota–miRNA axis may also be used as a therapeutic approach for patients with COVID-19 [32]. Recently, in 2021, it was shown that there is a link between the use of probiotics, omega-3 fatty acids, multivitamin and vitamin D supplements, and reduce the risk of testing positive for SARS-CoV-2 in the case of women (this observation was not confirmed in men) [38]. Probiotics contribute to the restoration of gut microbiota balance and functional homeostasis and prevent the invasion of pathogenic microbes through, among others, increasing the secretion of mucins [35,38,39,40]. Therefore, the aforementioned studies suggest that gut microbiota modification through the administration of probiotic strains appears to be much needed in COVID-19 patients. In the present narrative review, we present the immunomodulatory properties of probiotics and their association with COVID-19. Initial studies show that microbiota alterations are specific to COVID-19 infection and may be related to the severity of the disease [41].

## 2. Materials and Methods

The literature review was conducted by all authors. Studies published up to 13 September 2021 were included. The search words were: (“coronavirus” OR “severe acute respiratory syndrome coronavirus 2” OR “severe acute respiratory syndrome coronavirus 2” OR “sars cov 2” OR “SARS-CoV-2” OR “COVID-19” OR “severe acute respiratory syndrome coronavirus 2”) AND (“Nutrition” OR “Diet” OR ”Home nutrition” OR “Enteral nutrition” OR “parenteral nutrition” OR “probiotics” OR “probiotic mixtures”). The inclusion criteria were articles published up to 13 September 2021 containing the search words listed above. The exclusion criteria were articles published after this date and/or not containing the above-mentioned search words and/or physical therapy associated with probiotic ingestion and/or prebiotic ingestion.

### 2.1. Nutritional Support and Nutritional Status of Patients with SARS-CoV-2 Infection

#### 2.1.1. Nutritional Status

The main nutritional consequences of COVID-19 include reduced food intake, hypercatabolism, and rapid muscle wasting. Several factors contribute to reduced food intake, such as dyspnea, anorexia, and stress [42]. Furthermore, gastrointestinal symptoms, for instance nausea, diarrhea, and abdominal pain/discomfort, also contribute to the worsening of patients’ nutritional status [43]. Notably, most patients who are admitted to the intensive care unit (ICU) are at high risk of malnutrition. Therefore, the nutritional status of COVID-19 patients should be assessed regularly. The assessment of nutritional status should also take into consideration the possibility of sarcopenia development [42]. Zhou et al. have noted that COVID-19 is related to negative outcomes in elderly and hypoalbuminemic patients [44]. Albumin is a negative acute-phase protein. The decrease of its concentration in blood serum not only depends on the presence of malnutrition, but it is also associated with inflammation [45]. The reduction of prealbumin levels may be related to the prediction of acute respiratory distress syndrome progression. Lymphopenia is also a negative prognostic factor in these patients [44].

As mentioned above, according to recently published studies, the outcome of patients with SARS-CoV-2 is strongly associated with their nutritional status [46,47]. In the study by Hyoung Im et al., the nutritional status of 50 hospitalized COVID-19 patients regarding vitamins (B1, B6, B12, D) and minerals (selenium, folate, zinc) was measured [18]. The deficiency of vitamin D was observed in 76% of patients (severe deficiency—a cut-off of ≤10 ng/dL—24% of patients in the COVID-19 group and 7.3% in the control group). Additionally, low selenium was noted in 42% of patients. No significant deficiencies were observed for the other measured parameters. However, it was noted that all severely ill patients had a deficiency of more than one nutrient. Overall, these results indicate that a deficiency of both vitamin D and selenium is associated with weakened immunity in COVID-19 patients. The authors also suggest that nutritional deficiencies may favor the onset of this disease and increase its severity [18]. Notably, vitamin D, which plays a significant role in human immunity, decreases acute respiratory infections and pneumonia [48,49]. Alexander et al. recommend early nutritional interventions with nutrients (i.e., zinc, selenium, and vitamin D) to raise the antiviral resistance against COVID-19 [50]. These supplementations should be introduced particularly in areas with a high risk of COVID-19 development or as soon as possible when suspecting a SARS-CoV-2 infection [50].

It is recommended to assess patients’ nutritional status using appropriate tools, for instance Nutritional Risk Screening 2002 (NRS 2002) [51]. Moreover, nutritional evaluation should be based on new criteria established for the diagnosis of malnutrition, i.e., Global Leadership Initiative on Malnutrition 2019 (GLIM 2019) [42]. These criteria may be very useful due to the fact that they take into account many nutritional aspects and allow assessing the severity of malnutrition. Furthermore, GLIM 2019 criteria are based on a global consensus and should thus be well suited to the COVID-19 epidemic [42].

#### 2.1.2. Nutrition Support

The introduction of appropriate nutrition support in patients suffering from COVID-19 depends on their nutritional status and disease severity. The oral diet is preferred in the case of confirmed mild COVID-19 infections. Additionally, it is preferred to provide nutritional counseling, monitor oral feeding, and introduce oral nutritional supplements (ONS) if necessary [51]. The oral high-calorie and high-protein diet enriched with ONS is preferred if patients can be fed effectively orally [52].

It should be emphasized that EN is preferred over PN in COVID-19 patients due to the promotion of gut integrity and immune functions [42,52]. EN should be introduced early after the admission of patients with SARS-CoV-2 to the ICU. Due to the fact that EN may cause adverse events, it is recommended to start enteral formula administration with trophic doses and increase dosage systematically, taking into account patient tolerance, hemodynamic stability, and metabolic aspects [53]. Currently, it is recommended to use the standard isomolar polymeric formula in this case. The average energy and protein administration should be around 25 kcal/kg/d and 1.3 g/kg/d, respectively [42,53]. Nevertheless, the EN should be started carefully due to the high risk of refeeding syndrome. Therefore, the increase of enteral formula should be conducted via the following schedule: Day I, 10 kcal/kg/d; Day II, 15 kcal/kg/d; Day III, 20 kcal/kg/d; Day 4, 25 kcal/kg/d [42]. Additionally, in the case of acute respiratory distress syndrome (ARDS), the enrichment of EN with omega-3 fatty acids (at 3.5 g of eicosapentaenoic acid and docosahexaenoic acid) is currently recommended [42].

PN should be initiated in the case of a high risk of aspiration, enteral formula intolerance, insufficient enteral feeding, and when the gut is not functioning effectively [42,53]. The introduction of PN should be considered if the energy and protein requirements, i.e., <70% of the needs, are not covered enterally. Supplemental PN should not be applied before Day 4 [42]. The parenteral formula (similarly to EN) may be enriched with omega-3 fatty acids [42].

Summarizing the aspects of nutritional status and nutrition support in COVID-19 patients, it should be emphasized that all of the patients should be considered at risk of malnutrition. The evaluation of their nutritional status should be conducted using standardized methods. The GLIM 2019 criteria can be useful in the context of the epidemic. Nutrition support, such as EN and particularly PN, requires careful assessment before introduction due to the high risk of refeeding syndrome. The type of nutrition support should be based on patients’ nutritional status and the severity of COVID-19 disease.

### 2.2. Patient Nutrition in Mild Course of SARS-CoV-2 Infection at Home

While there are no strict guidelines in this regard, it appears that nutrition should be built (just like in the case of a more severe course) around a balanced diet, containing the optimal amounts of proteins and other nutrients and providing adequate amounts of vitamins and macro- and microelements [52].

In the course of the disease, it is essential that the patient should maintain all the metabolic functions, normal body weight, and muscle mass, hence a high-calorie, high-protein diet is recommended. The daily calorie intake should amount to ≥1500–2000, with 75–100 g/d of protein. If the patient has no appetite, they should be provided within 24–48 h with medical nutrition products, also known as foods for special medical purposes or oral nutrition supplements (ONS), to satisfy their nutritional requirements. Nutridrinks should be administered for ≥1 month, providing ≥400 kcal per day and ≥30 g protein per day [52,54,55].

## 3. Results

### 3.1. Diet as a Factor in Boosting the Immune System

Research has shown that vitamin and mineral deficiencies have an adverse effect on the efficacy of treatment in the course of viral infections [56]. Nutritional supplements enhance immunity [57]. Micronutrients that are of particular importance in patient care include omega-3 fatty acids (EPA and DHA), selenium and zinc, and vitamins A, E, D, C, B6, and B12. Notably, vitamin C, D, zinc, selenium, and others inhibit the production of inflammatory mediators during viral infection [56]. In a recent paper, Khatiwada and Subedi described selenium as a powerful immune factor in the fight against SARS-CoV-2, at the same time emphasizing the role of other micronutrients mentioned above [58]. The ESPEN guidelines affirm the positive role of these immunostimulating food components while observing that, with few available studies, it is difficult to recommend dosage in COVID-19 patients. It would appear that dietary intake of these nutrients in patients should be increased to well beyond the recommended dietary allowances (RDAs). For more information, please refer to the papers by Di Rezno et al. and Calder et al. [58,59].

#### 3.1.1. Vitamin D

Vitamin D has immunostimulating effects due to the presence of the vitamin D receptor in monocytes, macrophages, and dendritic cells. Additionally, it also stimulates the production of antimicrobial proteins, such as cathelicidin and β-defensin. By participating in their synthesis, vitamin D promotes the normal composition of intestinal microbiota, upregulating the expression of tight junction proteins in intestinal epithelial cells. Its major sources include oily fish, liver, eggs, and wild mushrooms [60]. The prophylactic dose of vitamin D amounts to 2000 IU/d (50 µg/d), 1600–4000 IU in obese adults, and 1200–2000 IU in obese children and adolescents. The dose of 20,000–50,000 IU is suggested to be supportive in respiratory tract infections [61,62].

#### 3.1.2. Vitamin A

Vitamin A is regarded as one of the most important micronutrients. Its principal role is in maintaining a mechanical barrier against pathogens by participating in the differentiation of epithelial tissue and skin. In the form of retinoic acid, it participates in the differentiation and proliferation of immune cells [63]. Additionally, it enhances the intestinal immune response. In the form of provitamin A, it can be found in vegetables, such as carrot, parsley leaf, spinach, kale, and broccoli, as well as fruit, notably apricot and peach. Beta-carotene is present in small quantities in milk and milk products such as butter. The daily prophylactic dose and that suggested in support of infections amounts to 900–700 µg/d [61].

#### 3.1.3. Vitamin C

Vitamin C is an essential water-soluble antioxidant, protecting immune cells against oxidative stress. Moreover, it is involved in cell signaling and epigenetic processes [63]. It has been associated with a shorter duration and less severe symptoms of upper and lower respiratory tract infections, at the same time lowering the risk of infection. Requirements for vitamin C are increased during infection, and its deficiency impairs phagocytosis [63]. Foods rich in vitamin C include parsley leaf, bell peppers, cabbage and relatives (*Brassica*), and citrus fruit [60]. The prophylactic dose amounts to 200 mg/d, and the supportive dose in respiratory tract infections is 1–2 g/d [61,62].

#### 3.1.4. Vitamin E

Vitamin E exists mainly in the form of tocopherols abundantly present in nuts and vegetable oils. In the form of tocotrienols, it can be found in some seeds and grains. In a lipid milieu, it acts as an antioxidant. It protects cell membranes from reactive oxygen species, supports the integrity of epithelial barriers, and stimulates T cells [63]. It is particularly important because of its role in regulating the body’s susceptibility to infectious diseases, including respiratory tract infections [63,64]. It reduces the risk of infection in healthy adults aged over 60 years. The prophylactic dose in men amounts to 10 mg and in women 8–11 mg of α-tocopherol equivalent/person/d. The dose suggested in support of respiratory tract infections is 135 mg/d [61].

#### 3.1.5. Zinc

Zinc deficiency, especially in children, is associated with an increased risk of diarrhea and development of pneumonia. It is particularly important for T cells, as it is needed for their maturation [63]. The dietary sources of zinc include shellfish, meat, liver, cheese, some cereals (buckwheat), and wholegrain bread [60]. The daily prophylactic dose is 8–11 mg/d, and in the course of respiratory tract infections, 30–50 mg/d is suggested [61].

#### 3.1.6. Selenium

Selenium is a nutrient of key importance for the immune system. It plays an important antioxidant role, affects leukocyte and natural killer (NK) cell function, and consequently modulates the antioxidant host defense system. Selenium is found in fish, shellfish, meat, eggs, and some nuts, notably Brazil nuts [58]. Currently, the recommended selenium intake for adults ranges between 25 and 100 μg/d, averaging at 60 μg/d for men and 53 μg/d for women. The tolerable upper intake level was established at 300–450 μg/d [61].

#### 3.1.7. Omega-3 Fatty Acids

The amount of omega-3 fatty acids recommended for prophylaxis is approximately 250 g/d, and in the case of infection, the suggested intake is 2–4 g/d [61]. Their main health benefits are attributable to their anti-inflammatory and antiplatelet effects due to their conversion in the body to active metabolites, namely resolvins and protectins [65]. They include linolenic acid (ALA), found primarily in seed oils, notably linseed oil, flaxseed, and chia. ALA is converted into other polyunsaturated acids: eicosapentaenoic acid (EPA) and docosahexaenoic acid (DHA). ALA cannot replace fish oils, in spite of being an EPA and DHA precursor, because of the low rate of conversion, which is inadequate to satisfy human nutritional requirements for those acids [66].

EPA and DHA are readily available from oily fish, algae, and seafood. Please note, however, that their content differs depending on fish species (Table 1). High amounts can be found in cold-water fatty fish, including such species as salmon, anchovies, herring, mackerel (Atlantic and Pacific), tuna (bluefin tuna and albacore), and sardines, with one of the highest levels of beneficial fatty acids. Algae are another good source, e.g., *Chlorella vulgaris* (CV) and *Fucus* sp. [67]. On the other hand, there are practically no omega-3 fatty acids in shrimp, lobster, scallop, tilapia, or cod.

Omega-3 fatty acids can also be supplied in the form of (fish oil) supplements, although according to the literature data, the majority do not provide satisfactory amounts of these valuable oils while containing contaminants dissolved in oil (Table 2) [68]. Most supplements labeled as fish oils contain as little as 12% DHA, 18% EPA, and more than 70% vegetable oil. On the other hand, fish oils with medicine status, e.g., Omacor, contain more than 40% DHA, more than 50% EPA, and just 6% of other omega-3 fatty acids [68]. Other recommendable omega-3 fatty acid preparations with medicine status include Omtryg, Epanova, and Vascepa.

#### 3.1.8. Curcumin

Curcumin is a natural therapeutic option. In a randomized trial (ClinicalTrials.gov identifier: CTRI/2020/05/025482), it was shown that oral curcumin (dose of 525 mg twice daily) with piperine (2.5 mg) may be used as adjuvant therapy for patients with COVID-19 [69].

### 3.2. Elements of Immunonutrition as a Potential Adjunct Therapy for COVID-19 Patients

In the context of immunonutrition, it is worth considering polyphenols. They are colorful substances with antioxidant [70] and anti-inflammatory properties [71,72] and documented effects on the condition of gut microbiota [73]. Good sources of polyphenols are cocoa products, darkly colored berries, seeds (for instance flaxseed), nuts (such as hazelnut, chestnut), vegetables, various species, and dried herbs [74]. Research has confirmed their positive participation in metabolic processes. For instance, they affect lipid metabolism, through inter alia reducing LDL cholesterol, increasing HDL cholesterol, and normalizing dyslipidemia [75]. Additionally, they activate the anti-inflammatory Nrf2 pathway [76] and inhibit NF-κB and AP-1-mediated proinflammatory cascades. It is important to note their beneficial role in the prevention of influenza infections [77,78]. It has been documented that polyphenols are capable of modifying cell signaling pathways and reducing viral replication [71,78]. A study carried out by Lin et al. investigated polyphenols found in grapes (*Vitis vinifera*), Huzhang (Japanese knotweed, *Polygonum cuspidatum*), and cranberry (*Vaccinium macrocarpon*) and found that they inhibited the replication of another virus, MERS, in an in vitro model [79].

Immunostimulating effects have also been observed in amino acids recognized as immunonutrients: glutamine and arginine [58,80,81]. Arginine regulates immune function and impacts metabolism. According to the literature data, its optimal intake is crucial when fighting inflammation in infections, fibrotic diseases, and immune regulation in general, seeing as it supports immune cell metabolism [58]. In the case of infection, the recommended intake is 25–35 g/d [82,83]. Glutamine, on the other hand, is the most versatile amino acid in our body. Glutamine consumption by immune cells is on par with that of glucose. In addition, it is an essential element in lymphocyte proliferation and neutrophil activation. Its depletion impairs immune function and increases susceptibility to infectious diseases [58].

### 3.3. Microbiota in COVID-19

The human gastrointestinal tract is inhabited by 10^14^ microorganisms, including bacteria, viruses, fungi, and archaea [84]. There are four other major phyla: *Proteobacteria*, *Fusobacteria*, *Actinobacteria*, and *Verrucomicrobia* [85,86,87].

Microbiota has a number of functions that are vital to human health and life. Gut bacteria participate in digestion and play an important role in healthy fatty acid metabolism. They also produce certain health-promoting nutrients, notably valuable short-chain fatty acids (SCFAs), vitamins, and amino acids [88]. The gt microbiome modulates the immune system and affects the metabolism of xenobiotics [89]. Moreover, commensal bacteria compete with other microorganisms for space and resources and produce antimicrobial proteins (AMPs), preventing the adhesion of pathogens and promoting intestinal barrier integrity [90,91]. Short-term modifications of eating habits have a negligible effect on bacterial composition, unlike antibiotic treatments, the impact of which is powerful, resulting in loss of taxonomic diversity and increased counts of potentially pathogenic bacteria [92]. This may lead to a state of dysbiosis and increased intestinal permeability [93].

Dysbiosis affects patients suffering from inflammatory bowel disease (IBD), irritable bowel syndrome (IBS), celiac disease, colorectal cancer, hepatic encephalopathy, and neurodegenerative diseases [94]. Over time, adverse changes in microbiota composition have also been confirmed in patients with type 2 diabetes mellitus, cardiovascular disease, and depression [95,96,97].

However, the effects of dysbiosis are not limited to the visceral area, seeing as gastrointestinal dysbiosis has been associated with respiratory diseases of inflammatory origin, i.e., allergy, asthma, and chronic obstructive pulmonary disease [98,99]. Substances produced by microbiota in the gut, for instance SCFAs, stimulate elements of the immune system, which are transported to the lungs through the lymphatic system [100,101]. Another controlled study in mice demonstrated that viral lung infection leads to unfavorable alterations in gut microbiota composition, manifesting with an increase in *Bacteroidetes* and decrease in *Firmicutes* abundance [102]. Gut microbiota may be a promising target for viral infection therapy [103,104]. Notably, gut microbiota (depending on its composition and activity) may both promote or prevent viral infections by regulating the immune response [103].

Recently, another item could be added to the list of factors affecting gut health, namely the SARS-CoV-2 pandemic. Intestinal disturbances observed in a severe course of SARS-CoV-2 infection are associated with gut dysbiosis. Hospitalized COVID-19 patients present with nausea, vomiting, loss of appetite, and diarrhea [105]. Notably, diarrhea affected from 1.3% to 29.3% of patients hospitalized due to a SARS-CoV-2 infection [105]. What is more, it was noted that patients with severe COVID-19 were more likely to manifest gastrointestinal symptoms than those with a milder course [106]. It has been proposed that diarrhea in these patients can be attributed to altered function of the ACE2 receptor due to the interaction between SARS-CoV-2 and ACE2 in the intestine [107]. ACE2 is present in many organs, including lung tissue, nasopharyngeal mucosa, brain, stomach epithelium, duodenum, small intestine, and colon, which renders the small and large intestine highly susceptible to SARS-CoV-2 infection [108,109,110]. ACE2 is a significant part of the renin–angiotensin system [111]. SARS-CoV-2 is mainly replicated in respiratory epithelial cells. It can infect immune cells, such as macrophages, monocytes, activated T cells, and dendritic cells [111]. SARS-CoV-2 uses the ACE2 receptor to enter cells [112]. After ACE2 binding, the virus enters host cell cytosol through cleavage of S protein [113]. This protein is one of the four structural proteins located on the outer surface of this virus, and it regulates the interactions of the virus with the host cell-surface receptors. It should be emphasized that SARS-CoV-2 shares 80% of its genome with other types of human coronavirus [113]. Recently, it was shown that SARS-CoV-2 and SARS-CoV use different regions in the S protein receptor-binding motif and present different interactions for binding ACE2 [113].

Both the incidence of diarrhea and high mortality rates among the elderly due to COVID-19 can be related to gut dysbiosis and the concept of the gut–lung axis [114]. As a result of adverse alterations in gut microbiota composition and impaired integrity of the intestinal epithelium, intestinal immune responses are upregulated, subsequently leading to the development of intestinal inflammation. This phenomenon is related to the already mentioned increased intestinal permeability, colloquially referred to as “leaky gut”, allowing for the passage of bacterial metabolites and bacteria to enter the bloodstream [114].

Currently, it is already known that lungs, like intestines, have their own microbial community, the function of which is to maintain respiratory system homeostasis, while a state of dysbiosis in this organ may confer susceptibility to lung disorders [115,116]. What is more, adverse alterations in respiratory microbiota also have a detrimental effect on gut microbiota. According to the scientific literature, viral infections of the respiratory tract decrease the relative counts of *Lactobacilli* and *Lactococci*, simultaneously increasing *Enterobacteriaceae* within the intestinal microbiota [117,118]. In COVID-19 patients, gut microbiota was found to be disturbed even in the absence of symptoms and following recovery [119]. Gu et al. noted that SARS-CoV-2 infection was associated with a marked decline in intestinal bacterial diversity and a significant increase in opportunistic pathogens: *Rothia*, *Actinomyces*, *Veillonella*, and *Streptococcus* [120]. Following the report from the study by Trompette et al., we now know that healthy gut microbiota supports immunity of the respiratory tract due to the fermentation of plant fiber and production of SCFAs, which may increase the number of dendritic cells in pulmonary tissue, thus dampening allergic responses, while low SCFA levels are associated with higher rates of allergic airway diseases [101]. Moreover, dysbiosis may allow the migration of SARS-CoV-2 from the lungs to the intestinal epithelium cells via the ACE2 receptor. It has been suggested that intestinal microbiota disorders may be of key importance in the recovery of patients with severe COVID-19 [41,114]. In a pilot study, significant adverse alterations in the composition of gut microbiota, which persisted longer than the symptoms of COVID-19 and SARS-CoV-2 infection were observed. Patients presented with higher amounts of *Coprobacillus*, *Clostridium hathewayi*, and *Clostridium ramosum* compared to the control group, while the abundance of the anti-inflammatory bacterium *Faecalibacterium prausnitzii* declined in relation to the increasing disease severity [41]. Interestingly, there is an inverse correlation between the strains of *Bacteroides dorei*, *Bacteroides ovatus*, *Bacteroides thetaiotaomicron,* and *Bacteroides massiliensis,* which downregulate ACE2 expression in murine gut, and SARS-CoV-2 levels in fecal samples from hospitalized patients [41]. It appears that the use of appropriate microbes inhibiting the activity of the ACE2 receptor may help reduce viral invasion of human cells and thus dampen the inflammatory process [26]. For instance, *Lactobacillus* and *Bifidobacterium* spp. synthesize proteins that are ACE inhibitors, with a potential for downgrading the severity of inflammation in patients with respiratory tract infections [121]. Meanwhile, there have been reports of decreasing abundance of those bacteria in patients infected with the SARS-CoV-2 virus [122,123]. No specific reason has been established for gut microbiota disturbances in COVID-19 patients, and it is uncertain whether they are caused by antiviral pharmacotherapy, the combination of disease symptoms, or the infection itself [124]. It would appear that another reason why severe COVID-19 is dangerous for patients may lie in the significant loss of appetite. A certain relationship has been observed between appetite loss and calorie reduction in mice and a significant increase in the ratio of *Bacteroidetes* to *Firmicutes* abundance, which was also noted in a study of microbiota alterations during respiratory viral infection [103,125]. The reduction of energy intake from food (which is observed in COVID-19 patients due to significant loss of appetite) may contribute to worsening their condition [126].

Apart from the viral infection, which may exacerbate the condition of COVID-19 patients by inducing intestinal inflammation and dysbiosis, scholars also mention antibiotic therapies, which are in widespread use in Western countries. Antibiotics taken for therapeutic purposes affect all bacteria, increasing the risk of gut colonization by pathogenic biota, consequently damaging the intestinal barrier. There are many publications discussing the adverse dysbiotic effects of antibiotics [92,127,128,129,130].

The academic literature exploring the role of probiotics in inhibiting the activity of viruses suggests there is real potential for the use of certain bacterial strains to improve survival rates in infected patients. However, our current knowledge of the specific mechanisms underlying the antiviral activity of bacteria remains uncertain. According to the FAO/WHO definition, probiotics are live microorganisms that, when administered in adequate amounts, provide health benefits to the host [131]. It has been hypothesized that the mechanisms may include the production of metabolites, bacteriocins, and antiviral compounds, hindering the adsorption and cell internalization of the virus and immunomodulation enhancing immunity to the virus [132]. The COVID-19 infection affects lung tissue then activates an inflammatory response, increasing the release of proinflammatory mediators, such as IFN-γ and TNF-α. Probiotics may regulate immune response in the respiratory system [133]. The immunostimulating properties of probiotics are determined by their components, i.e., cytokine-activating substances such as lipoteichoic acid, peptidoglycan, Toll-like receptor (TLR) ligands, and muramyl dipeptide [134].

Probiotics may support patients by inhibiting the ACE2 receptor, i.e., the passage of the virus into the cell, and may also be effective in suppressing the immune response caused by the proinflammatory cytokine cascade [134]. Patients with COVID-19 have elevated levels of various cytokines such as IL-2, IL-4, IL-6, and IL-10, TNF-α, IFN-γ and may present with the so-called cytokine storm [135]. Moreover, it has been observed that elevated levels of these cytokines in patients represent a risk factor for a severe disease course and may require hospitalization in an ICU [135]. It is worth noting that *Lactobacillus casei* demonstrates stimulating effects on the cells of gut-associated lymph tissue (GALT) to produce immunoglobulin A (IgA). Pediatric patients with diarrhea received 2 × 10^8^ colony-forming units (CFU)/250 mg orally twice daily for seven days by intravenous fluid hydration [136]. Secretory IgA is recognized as one of the most crucial components of innate immunity during infection [137]. Intestinal colonization by probiotic bacteria results in higher B cell counts and, consequently, increased IgA expression in lymph nodes and colon, as well as higher counts of Th cells and dendritic cells, responsible for IL-23 expression. These reactions decrease the occurrence of respiratory viral infections and attenuate their symptoms [135]. Interestingly, *Lactobacillus paracasei* contains ACE2 [138]. Oral administration of ACE2 in *Lactobacillus paracasei* could serve as protection against COVID-19 by binding SARS-CoV-2 and thus preventing its interaction with ACE2 receptors in human cells and the resulting infection [139]. In a systematic review and meta-analysis of randomized controlled trials, it was suggested that probiotics and prebiotics enhance immunogenicity. They impact seroconversion and seroprotection rates in influenza-vaccinated adults [140]. Kullar et al. showed that a multistrain probiotic mixture may be effective in reducing the incidence of diarrhea related to COVID-19 [141]. However, this was confirmed in one trial. The authors reviewed the literature up to March 2021. Additionally, a study registered in the ClinicalTrials.gov system (Identifier: NCT04517422) assessed the usage of a probiotic mixture, including *Lactobacillus plantarum* CECT7481, *Lactobacillus plantarum* CECT 7484, *Lactobacillus plantarum* CECT 7485, and *P. acidilactici* CECT 7483, in COVID-19 patients. The current status of this trial is “completed” as of 06/05/2021, and the results will be published soon.

A range of different probiotics has been examined to date in terms of their health benefits in respiratory diseases [142]. The most prominent among them, lactic acid bacteria (LAB) *Lactobacillus* spp. and *Bifidobacteria*, have been studied most extensively in this regard. The largest number of studies on the health effects of LAB investigated their activity in viral infections with H1N1 and RSV. It was demonstrated that *Lactobacillus plantarum L-137*, *L. plantarum DK119, L. rhamnosus CLR1505, L. gasseri* TMC0356, *Bifidobacterium longum BB536*, and *B. animalis* ssp. *Lactis BB12* participates in antiviral defense, modulating immune response and cytokine production. Moreover, all of the above-mentioned probiotics shortened the duration and reduced the severity of infections and improved intestinal health and overall immunity [143]. The results of randomized double-blind placebo-controlled studies in humans show a beneficial effect of probiotic therapy on respiratory tract infections (RTIs). Administration of *L. rhamnosus* GG (10^8^ CFUs) ingested in milk 3× daily for 7 months, a combination of *L. acidophilus* (min. 10^9^/capsule) and *B. bifidum* (min. 10^9^/capsule) (strain information not provided) in capsules 2× daily for 3 months, and a mixture of *L. acidophilus* (min. 10^9^/capsule) and *B. bifidum* (min. 10^9^/capsule) (strain information not provided) in capsules 2× daily or *L. casei* DN 114001 2 × daily in fermented yogurt were associated with a reduced risk of RTIs and duration of RTI episodes, reduced cough, risk of fever, and rhinorrhea in children and adults [132]. Antiviral action was also demonstrated in *Lactobacillus fermentum* ACA-DC179, *Enterococcus faecium* PCK38, *Lactobacillus pentosus* PCA227, and *Lactobacillus plantarum* PCA236 and PCS22 [144]. Please note, however, that gut microbiota is much more diverse and is not limited to the bacterial genera *Bifidobacteria* and *Lactobacillus* spp. For this reason, more research should be available on other types of probiotic bacteria in order to provide for the optimal prevention and treatment of viral infections, including those caused by SARS-CoV-2 [142]. Moreover, the appropriate doses of particular probiotics in these cases are not yet well established. Due to the fact that the administration of probiotics in COVID-19 is a relatively undiscovered field, there is a need to conduct further studies, also addressing the potential adverse events of probiotic consumption.

As mentioned above, the gut microbiota is strongly linked to the functioning of the immune system. Some respiratory infections are related to dysbiotic alterations of gut microbiota. Hegazy et al. investigated the role of nutritional and lifestyle habits in the context of modulation of gut microbiota and COVID-19 outcomes [145]. This study included 200 COVID-19 patients. It was noted that daily consumption of foods containing prebiotics, as well as less sugar, regular physical activity, and adequate sleep, is associated with milder disease and rapid clearance of the virus. The authors concluded that the aforementioned factors can potentially modulate the gut microbiome, and the severity of the disease may be reduced [145].

## 4. Conclusions

In conclusion, the treatment of gut dysbiosis involving an adequate intake of prebiotic dietary fiber and probiotics could turn out to be an immensely helpful instrument for immunomodulation, both in COVID-19 patients and prophylactically in individuals with no history of infection [143,146]. A balanced diet implemented at the right moment is a powerful tool supporting the patient’s body. The use of a probiotic/prebiotic therapy in patients with diarrhea appears to be a beneficial adjunct therapy (Figure 1). The use of specific probiotics demonstrating potential for reducing viral pathogenicity and severity of symptoms caused by SARS-CoV-2 may provide significant support for patients, but detailed guidelines need to be developed [147]. Without a doubt, an increased intake of proteins, vitamins, and micronutrients will promote immune function in those who have been infected. In addition, anti-inflammatory dietary components such as fatty acids EPA and DHA and polyphenols are capable of alleviating the proinflammatory effects of the virus.

## Figures and Tables

**Figure 1 nutrients-13-03385-f001:**
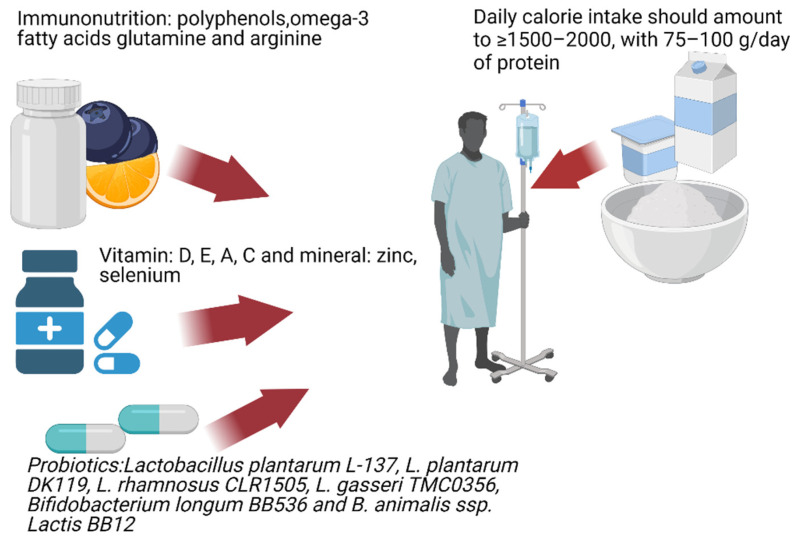
Immunonutrition patients with COVID-19.

**Table 1 nutrients-13-03385-t001:** In order to supply 250 mg/d of EPA and DHA (for prophylactic purposes), one should eat [54].

Scheme 1	Quantity (g/d)	Amount Supplied	Frequency	Omega-3 Fatty Acids
Mackerel pâté	100 g	4.7 g	Daily	EPA + DHA
Salmon and sardines	142 g	4.5 g	Daily	EPA + DHA
Baked herring	150 g	1.2 g	3 days/week	EPA + DHA
Sea trout	150 g	3.2 g	Daily	EPA + DHA
Cod	150 g	0.3 g	Daily	EPA + DHA

EPA, eicosapentaenoic acid; DHA, docosahexaenoic acid.

**Table 2 nutrients-13-03385-t002:** EPA and DHA content of commonly used supplements [54].

Supplement	Content	Supplement
Cod liver oil	200 mg EPA and DHA	More EPA than DHA, not medicine grade
Standard fish oil	300 mg EPA and DHA	More EPA than DHA, not medicine grade
Concentrated fish oil	450–600 mg EPA and DHA	More EPA than DHA, not medicine grade
Algae oil	400 mg	Mainly DHA, not medicine grade
Linseed oil	No information	Mainly alpha-linolenic acid, not medicine grade
Omacor	460 mg EPA and 380 mg DHA	medicine grade

## Data Availability

Not applicable.

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
