# Peer review of "Patient Nutrition and Probiotic Therapy in COVID-19: What Do We Know in 2021?"

_nutrients, 2021, doi:10.3390/nu13103385_

Round 1

Reviewer 1 Report

revisions are great. Good to go!

Author Response

Reviewer 1

Revisions are great. Good to go!

Thank you for nice comment. We hope our paper will improve the current knowledge about COVID-19.

Reviewer 2 Report

The submitted ms is interesting and aimed to review current data and ideas about the use of probiotics as supportive therapy in COVID-19 patients.

This is a topic of great interest, and a mass of data are currently available about the utility of probiotics in different infectious diseases, including COVID-19.

However, I've some concerns about this manuscript.

Major concerns:

First of all, it is quite outdated about COVID-19 epidemiology and treatment. In fact, the data in Introduction are as on 6 April, 2021 and the same for the related references (# 1 and 4). It is confirmed in Matherials and Methods, where  the authors state that "Studies published up to15/04/2021 were included". Data and related references must be updated to make this paper useful for researchers and clinicians involved in facing COVID-19.

In M&M section, must be added databanks used for data collection. In addition, the authors should report the range of dates for their search, not only the end date (15/04/2021). This is imporant also to understand the rationale of probiotics use in infectious diseases,

For reccomended daily amount of vitamins, minerals, and other nutrients please add the main scientific source (scientific association? consensus paper? other?)

Regarding the list of best sources of polyphenols, it must be revised considering and citing the paper by Perez-Jimenez et al. (see below the suggested references)

The authors state repeatidley about SARS-CoV-2 action on ACE receptors and the immune response against the virus, but no informations about the molecular aspects of these processes (both for the virus and the host) are discussed.

Please read and discuss at least the following papers: PMID: 34442684, 34432408, 34401711, 34315200, 34308122, 34179059, 34162042, 34122090, 34100300, 34096487,  34054866, 34054314,  33484986, 33269412, 32574260, 21045839 

Minor concerns:

 Revise the text for English language and grammar, and for typos (e.g., SARS-CoV-2 is the correct name of the virus, not SARS-cov-2 (line 91) or Sars-CoV-2 virus (line 23).

Author Response

Reviewer 2

The submitted ms is interesting and aimed to review current data and ideas about the use of probiotics as supportive therapy in COVID-19 patients.

This is a topic of great interest, and a mass of data are currently available about the utility of probiotics in different infectious diseases, including COVID-19.

However, I've some concerns about this manuscript.

Major concerns:

First of all, it is quite outdated about COVID-19 epidemiology and treatment. In fact, the data in Introduction are as on 6 April, 2021 and the same for the related references (# 1 and 4). It is confirmed in Matherials and Methods, where  the authors state that "Studies published up to15/04/2021 were included". Data and related references must be updated to make this paper useful for researchers and clinicians involved in facing COVID-19.

In M&M section, must be added databanks used for data collection. In addition, the authors should report the range of dates for their search, not only the end date (15/04/2021). This is imporant also to understand the rationale of probiotics use in infectious diseases.

Thank you for these significant comments, which allowed us to improve the quality of this paper. According to your suggestions, we revised our manuscript through updating references, which were published until 13/09/2021. We discussed below recommended by you papers and others which were recently published.

For reccomended daily amount of vitamins, minerals, and other nutrients please add the main scientific source (scientific association? consensus paper? other?)

According to your suggestion, we cited papers which recommended these daily amounts of aforementioned elements.

Di Renzo, L.; Gualtieri, P.; Pivari, F.; Soldati, L.; Attinà, A.; Leggeri, C.; et al. COVID-19: Is there a role for immunonutrition in obese patient?. J Transl Med. 2020, 18, 415. doi: 10.1186/s12967-020-02594-4.

Bold, J.; Harris, M.; Fellows, L.; Chouchane, M. Nutrition, the digestive system and immunity in COVID-19 infection. Gastroenterol Hepatol Bed Bench. 2020, 13, 331-340. 

Regarding the list of best sources of polyphenols, it must be revised considering and citing the paper by Perez-Jimenez et al. (see below the suggested references)

Thank you for this comment. We corrected the list of the best sources of polyphenols. According to your suggestion, we discussed and cited following paper:

Pérez-Jiménez, J.; Neveu, V.; Vos, F.; Scalbert, A. Identification of the 100 richest dietary sources of polyphenols: an appli-cation of the Phenol-Explorer database. Eur J Clin Nutr. 2010, 64, 112-120. doi: 10.1038/ejcn.2010.221.

The authors state repeatidley about SARS-CoV-2 action on ACE receptors and the immune response against the virus, but no informations about the molecular aspects of these processes (both for the virus and the host) are discussed.

We developed our manuscript by discussing molecular aspects regarding SARS-CoV-2 and ACE receptors. We cited recently published data.

Please read and discuss at least the following papers: PMID: 34442684, 34432408, 34401711, 34315200, 34308122, 34179059, 34162042, 34122090, 34100300, 34096487,  34054866, 34054314,  33484986, 33269412, 32574260, 21045839 

Thank you for this comment which significantly allowed us to improve the quality of this paper. We discussed and cited all aforementioned articles.

Minor concerns:

 Revise the text for English language and grammar, and for typos (e.g., SARS-CoV-2 is the correct name of the virus, not SARS-cov-2 (line 91) or Sars-CoV-2 virus (line 23).

We corrected typos regarding the name of SARS-CoV-2. We also corrected our paper with English language to improve the readability of this manuscript.

Round 2

Reviewer 2 Report

Tha authors addrssed all the comments, fixing all the issues and improving the manuscript's quality.

I suggest to accept it in present form.

This manuscript is a resubmission of an earlier submission. The following is a list of the peer review reports and author responses from that submission.

Round 1

Reviewer 1 Report

nutrients-1291680

Title

Patient nutrition and probiotic therapy in COVID-19: what do we know in 2021?

Abstract

  1. Line 24. Authors should inform in the abstract if this is a narrative or systematic review.
  2. Line 28. “selen” should be selenium?
  3. Abstract results and conclusions should provide more specific information, such as doses, period of treatment and best protocols.

Introduction

  1. Line 74. EN should be spelled out.
  2. Line 74. UCI should be spelled out.
  3. Line 81. This sentence has a typo. Revise.
  4. Introduction is very well written. Authors reported several studies about the importance of nutritional surveillance after COVID-19. However, results from studies evaluating hospitalized and community-based samples are mixed. Besides, results regarding obese and malnourished individuals are also mixed. Another problem is that probiotic potential as a therapy for post-Covid patients is not explored at all. Probiotics only appeared in the last paragraph. Authors should explain physiological mechanisms that indicates potential benefits for probiotic ingestion as a preventive of therapeutical agent. It is important to state benefits for obese and for population under malnutrition conditions. Age, hospitalization period, comorbities, dimer D are also important in this context.

Material and Methods

  1. Authors should indicate databases and search strategies more clearly. Besides, authors should state inclusion and exclusion criteria.
  2. Authors present results of previous articles without differentiation levels of SARS-Cov-2 infection. Hospitalized patients are probably more prone to be deficient. In the same line, hospitalized patients tend to present more severe disease and consequently more prone to show nutritional problems.
  3. In Material and Methods section there is no mention to probiotics. The title of the article is not adequate for what authors describe until Introduction and Material and Methods Section.
  4. Subsections in Material and Methods make no sense. Why these subsections can not be part of the results section. Indeed, it is not clear to me the distribution of subsections between Methods and Results. The main part of the article (regarding its title) is only one section at the end of the article. Authors should rethink defining probiotics as the main part of the present article.
  5. Authors should try to state consistent evidence regarding study design and quality. Besides, doses and regimen should be discussed deeply. What about side effects? Should we add prebiotics to enhance probiotics effects? What are the most promising strategeies?

Author Response

Dear Sir or Madame,

Thank you for taking the time to review our article and all your helpful suggestions. All requested changes have been implemented in a revised version of the article and can be viewed as red color of text. We do hope our modifications will meet your expectations and standards.

For your convenience, all points risen in the review are addressed below in separate paragraphs.

Reviewer 2 Report

An interesting review of nutrition and probiotic therapies for possible role in COVID patients. As far as it goes, this reviews where we are on research, but the area of probiotics needs to be revised. The authors present mainly only animal data for some probiotics and how they reduce influenza infections. This is surprising, as they completely IGNORE the vast body of knowledge from human clinical trials. This is mis-leading. They need to include clinical trials of probiotics used against influenza and ventilator-associated pneumonia (a complication seen in COVID patients).  Also they failed to include a good review of this very topic. (Kullar R 2021 Antibiotics).

MINOR COMMENTS:
LINES 468 AND 481. Please use italics for genus and species names.

LInes 476-479. mIssing where references #115 and #116 should be linked with?

LInes 484-485. The line "A range of different probiotics..." needs a citation to support this statement. 

Author Response

(The authors gave the same response as above.)

Reviewer 3 Report

Dr Hawrylkowicz and colleagues provide a review on nutritional factors and Covid-19. They also explore the potential use of probiotics to modulate the risk of the disease. 

Overall, the manuscript is topical in view of the global significance of the disease.

Overall the manuscript is comprehensive. It would benefit from tidying up and shortening significantly. The infographic is a good snaphot of potential interventions. The basis for nutritional supplements is summarized extensively. As mentioned above, the section of probiotics should be shortened somewhat. 

Formal aspects:

-all bacterial genera and strains should be written according to current classification and convention throughout the manuscript

L 428: Western .. (capital )

table 2: suggest: table heading: EPA and DHA content of commonly used supplements

Author Response

(The authors gave the same response as above.)

Round 2

Reviewer 1 Report

I could not find a response letter for each point raised in the first review. It seems to me authors just altered the main document and all changes were highlight in blue font. If this is the case, authors should provide a response letter accordingly. For example, authors did not answered my question regarding the title.

Kind regards

Author Response

I am very sorry for this misunderstanding. When I sent the previous replies I pressed the submit button too quickly by mistake. I am sending a letter to the Editor-in-Chief to forward the letter to Rev 1 . But the letter apparently did not arrive. I am sending it correctly, sorry for the inconvenience 

Round 3

Reviewer 1 Report

Dear authors, 

I acknowledge your effort in answering each suggestion. Still, some of them are not yet adequate in my opinion, as follows:

1. Abstract results and conclusions should provide more specific information, such as doses, period of treatment and best protocols. In this suggestion, I referred to probiotic ingestion protocols, not daily requirements of micronutrients. Indeed, the added information is recommended for everyone, not only to COVID-patients.

2. Probiotic potential as a therapy for post-Covid patients is not explored at the introduction section. Authors should explain physiological mechanisms that indicates potential benefits for probiotic ingestion as a preventive of therapeutical agent. From the introduction I can not see any link to the title of the article.

3. Inclusion and exclusion criteria are not adequate. For example, articles investigating probiotic supplementation were included? What about prebiotic and probiotic mixtures were also included? Physical therapy associated to probiotic ingestion were excluded? Was there a time period of probiotic ingestion in order to be included?

4. I do not understand this answer: “In our manuscript we tried to note if studies regard hospitalized patients”.

5. I am still not convinced that probiotics is the main focus of this article. Indded authors made a broader evolution regarding nutrition and probiotics are just part of it.

6. Authors did not discuss doses and protocols. They barely mentioned species. Please, be more emphatic regarding protocols.